# Structural variants exhibit widespread allelic heterogeneity and shape variation in complex traits

Mahul Chakraborty [1*], J.J. Emerson [1], Stuart J. Macdonald [2] & Anthony D. Long [1*]

It has been hypothesized that individually-rare hidden structural variants (SVs) could account for a significant fraction of variation in complex traits. Here we identified more than 20,000 euchromatic SVs from 14 *Drosophila melanogaster* genome assemblies, of which ~40% are invisible to high specificity short-read genotyping approaches. SVs are common, with 31.5% of diploid individuals harboring a SV in genes larger than 5kb, and 24% harboring multiple SVs in genes larger than 10kb. SV minor allele frequencies are rarer than amino acid polymorphisms, suggesting that SVs are more deleterious. We show that a number of functionally important genes harbor previously hidden structural variants likely to affect complex phenotypes. Furthermore, SVs are overrepresented in candidate genes associated with quantitative trait loci mapped using the Drosophila Synthetic Population Resource. We conclude that SVs are ubiquitous, frequently constitute a heterogeneous allelic series, and can act as rare alleles of large effect.

[1] Department of Ecology and Evolutionary Biology, University of California Irvine, Irvine, CA 92697, USA. [2] Department of Molecular Biosciences, University of Kansas, Lawrence, KS 66045, USA. *email: mchakrab@uci.edu; tdlong@uci.edu

Understanding the molecular basis of heritable variation in complex traits is of central importance to evolution, animal and plant breeding, and medical genetics[1–4]. Over the last decade, short read genomic data (50–150 bp reads) appropriate for characterizing SNPs and small indels in non-repetitive genomic regions has accumulated at an exponential rate[5,6]. This in turn has catalyzed hundreds of quantitative trait locus (QTL) mapping and genome-wide association (GWAS) studies in model organisms, humans, and agriculturally important animals and plants[7–9]. Despite these efforts, for most traits, GWAS hits only explain a small fraction of known trait heritability[10,11]. One hypothesis accounting for hidden genetic variation is that individually rare hidden mutations that alter genome structure make significant contributions to complex trait variation[11,12]. These structural variants (SVs) change the genome via duplication, deletion, transposition, and inversion of sequences. This hypothesis is attractive since rare causative variants are difficult to detect with GWAS[13]. Moreover, genotyping approaches based on short reads or microarrays fail to detect a significant number of SVs[14,15]. Finally, it is reasonable to assume that SVs are on average likely to be more deleterious and deleterious more often than SNPs[16–19].

High quality genomes provide a direct and reliable path to comprehensive identification of SVs[15,20,21]. To achieve this goal, we assembled reference-quality genomes for fourteen geographically diverse *Drosophila melanogaster* strains (Fig. 1a) using single molecule real time sequencing[22]. These assemblies are contiguous and complete (N50 18.9–22.3 Mb; BUSCO[23] 99.9–100%) (Table 1, Fig. 1b, Supplementary Table 1), making them comparable to the *D. melanogaster* reference genome, arguably the best metazoan genome assembly. Thirteen of the fourteen strains are near isogenic founders of the Drosophila synthetic population resources (DSPR)[24], a large set of advanced intercross recombinant inbred lines (RILs) designed to map QTLs[25]. We also assembled the genome of Oregon-R, an outbred stock widely used as a "wild-type" strain both by Drosophila geneticists and by large scale community projects like mod-ENCODE[26–28].

Using these reference quality genome assemblies, we show that SVs are common in *D. melanogaster* genes, with almost one third of diploid individuals harboring an SV in genes larger than 5 kb, and more than a third of burdened genes carrying multiple SVs. The site frequency spectrum (SFS) of SV alleles relative to amino acid polymorphisms suggests that SVs are under stronger purifying selection, and thus are more likely to impact phenotype than nonsynonymous SNPs. We further show that a number of functionally important genes harbor previously hidden SVs likely to affect complex phenotypes (e.g., *Cyp6g1, Drsl5, Cyp28d1, Cyp28d2, InR,* and *Gss1&2*). Finally, we find that SVs are over-represented in candidate genes associated with mapped QTL. We conclude that SVs are pervasive in genomes, frequently manifest as heterogeneous allelic series affecting the same gene, and exhibit all the properties that make them prime candidates for being rare alleles of large effect.

## Results

**De novo assembly reveals novel functionally important SVs.** Our assemblies are extremely contiguous, with the majority of each chromosome arm represented by a single contig (Fig. 1b). We also close the two remaining gaps in the major chromosome arms of the euchromatic *D. melanogaster* reference genome[29] in all our assemblies (Supplementary Figs. 1–3). We identified SVs by comparing each assembly to the reference ISO1 genome[15], focusing our attention on large (>100 bp) euchromatic SVs (Supplementary Table 2), and ignoring heterochromatin regions

as they are gene poor[30] and require specialized assembly approaches and extensive validation[31]. Manual inspection of 267 randomly sampled SVs indicate that mis-annotations are rare (3/267), and occur in ambiguously aligned structurally complex genomic regions (Supplementary Fig. 5; see Methods). We discovered 7347 TE insertions, 1178 duplication CNVs, 4347 indels, and 62 inversions in the 94.5 Mb of euchromatin spanning the five major chromosome arms across the DSPR founders (Fig. 1c–d). Each founder strain exhibits 637 TE insertions, 134 duplications, 694 indels, and 7 inversions on average (Table 2). We estimate that 36% of non-reference TEs, 26% of deletions, 48% of insertions, 60% of duplication CNVs are not routinely detected using high coverage paired end Illumina reads and high specificity SV genotyping methods[15] (Supplementary Fig. 6)

We uncover many examples of previously hidden SVs predicted to affect complex traits. Extensive evidence links complex SV alleles of the cytochrome P450 gene *Cyp6g1* to varying levels of DDT resistance[32,33]. Despite extensive study of this locus, we discovered three new SV alleles involving TE insertions that likely have different functional consequences (Supplementary Fig. 7a, b). Similarly, we discovered a previously hidden tandem duplication of the antifungal, innate immunity gene *Drsl5*[34] that exhibits >1000-fold higher expression relative to its single copy counterpart in line A4 (Supplementary Fig. 8a, b). Read pair orientation and split read methods failed to detect this mutation because one allele bears a 5 kb spacer sequence derived from the first exon and intron of *Kst* inserted between the gene copies (Supplementary Fig. 8a). Another duplicate allele of *Drsl5* contains a *Tirant* LTR retrotransposon inserted into the same spacer sequence (Supplementary Fig. 8a). We also easily detect the two SV mutations underlying the *D. melanogaster* recessive visible genes *cinnabar*[35] (*cn*) and *speck* (*sp*) present in the ISO1 reference genome[36] (Supplementary Figs. 9 and 10). In the case of *sp* a large insertion in the reference genome is mis-annotated as an intron. For *cn* a large exonic deletion is not identified as such[36]. Both alleles are likely knock-outs.

**SVs are deleterious**. Most TEs and duplicates are present in only one strain (Fig. 1e), with the folded SFS of the TEs and duplicates exhibiting a greater proportion of rare variants than non-synonymous SNPs (nsSNPs) assayed in the same strains (Fig. 1e; $p$-value $< 1 \times 10^{-10}$, $\chi^2$ test between frequency classes of these two types of SVs and non-synonymous SNPs). Since nsSNPs were ascertained via high coverage short reads from virtually isogenic strains[24], the low frequency skew of the site frequency spectrum of SVs relative to nsSNPs is unlikely due to SNP miscalls (see Methods). It is well-known that the SFS is affected by demographic history[37,38], but selective constraints can be inferred with some confidence by comparing site classes from the same sample[37,39–41]. The skew toward rare variants we observe in our SVs relative to nsSNPs is strongly indicative of SVs being under stronger purifying selection, consistent with previous work in which SVs were ascertained with higher bias and/or errors[16,18,19]. Furthermore, TEs are more enriched for rare variants than duplicates, indicating that TE insertions as a class are more deleterious than duplicates (Fig. 1e; $p$-value $< 1 \times 10^{-10}$, $\chi^2$ test between frequency classes of TEs and duplicates). Under mutation selection balance models[42,43], rare deleterious variants (minor allele frequency or MAF <1%) are predicted to contribute significantly to the variation in complex traits, yet are unlikely to be tagged by SNPs typically used in GWAS experiments[10]. Although demography can impact the proportion of variation due to rare deleterious alleles, recent population bottlenecks or growth[44,45] tend to amplify the contribution of rare alleles to variation in a complex trait.

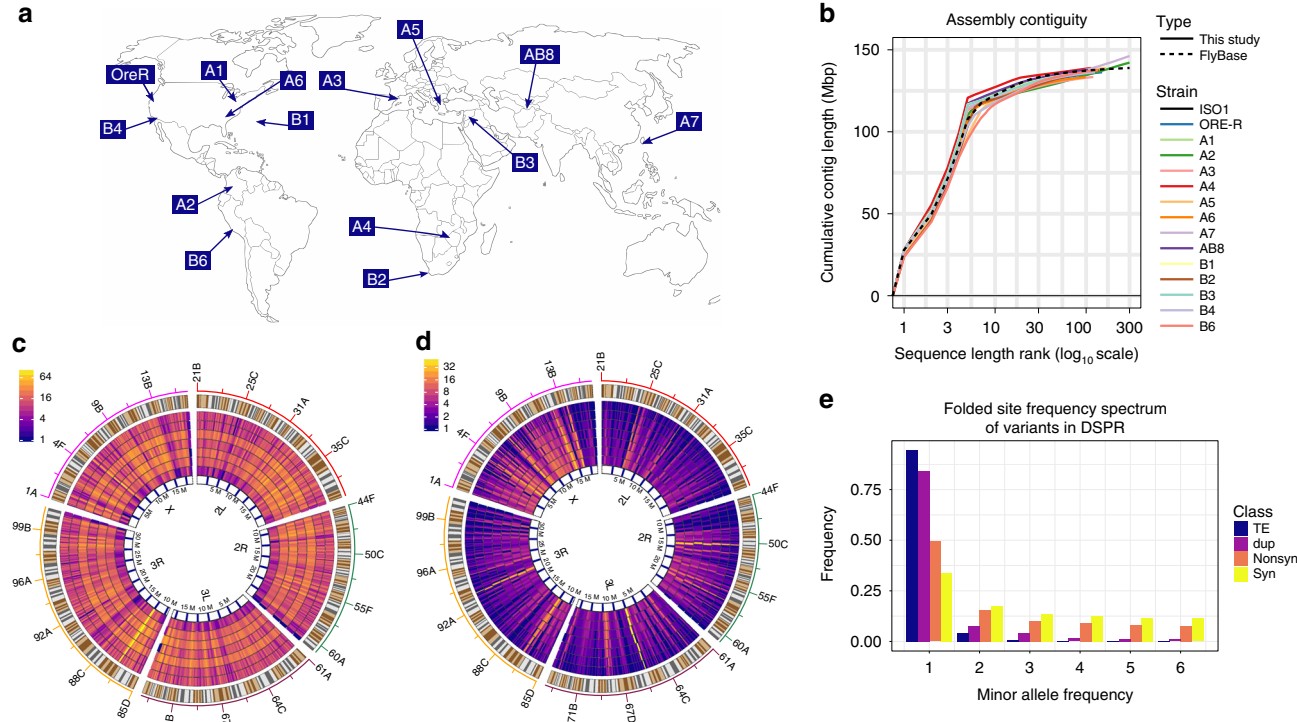

**Fig. 1** SVs in fourteen geographically diverse *D. melanogaster* strains. **a** Geographic locations of the sequenced strains of *D. melanogaster* (map source: www. outline-world-map.com). As shown here, the founder strains from DSPR and Oregon-R originate from diverse worldwide populations. **b** Cumulative contiguity plot showing comparison of assembly contiguity between the reference strain ISO1 and our 14 assemblies. **c** Distribution of euchromatic TE insertions across the major chromosome arms. The outermost track represents the chromosome ideogram, showing the locations of named bands. Each subsequent inner track shows distributions of TE insertions per genomic window of fixed size, ranging from 100–400 kb in 50 kb increments. Details of the TE rich region (yellow streak) on 3R (87B;12.47–12.5 Mb) is shown in Supplementary Fig. 4. **d** Distribution of duplication CNVs within euchromatin of major chromosome arms. The outermost track represents ideogram as in **c**. Inner tracks represent distributions of duplication CNVs in windows of varying sizes as in **c**. Unlike TEs, distribution of duplications are less uniform within and between the chromosomes. **e** Counts of minor allele frequency for TE, duplicated sequences (dup), nonsynonymous SNPs (Nonsyn), and synonymous SNPs (Syn)

**Table 1 Summary of assembly metrics**

| Strains | Assembly size (Mb) | Contig N50 (Mb) | # of scaffolds | Complete BUSCO (arthropoda) |
|---|---|---|---|---|
| ISO1 | 139.5 | 21.4 | 1856 | 100 |
| A1 | 137.6 | 21.8 | 76 | 100 |
| A2 | 142.4 | 22.3 | 193 | 99.9 |
| A3 | 133.3 | 21.6 | 44 | 100 |
| A4 | 139.6 | 22.4 | 95 | 100 |
| A5 | 138.9 | 20.9 | 99 | 99.9 |
| A6 | 133.3 | 21.5 | 29 | 100 |
| A7 | 146.8 | 21.5 | 263 | 100 |
| AB8 | 137.7 | 21.7 | 56 | 100 |
| B1 | 135.9 | 21.8 | 39 | 100 |
| B2 | 137.4 | 18.9 | 58 | 100 |
| B3 | 136.2 | 21.4 | 43 | 100 |
| B4 | 136.2 | 20 | 65 | 100 |
| B6 | 137.4 | 18.5 | 61 | 100 |
| Ore | 136.4 | 21.5 | 75 | 100 |

N50 = sequence length such that 50% of the assembly is contained within sequences of that length or longer
BUSCO = Benchmarking Universal Single Copy Orthologs

**Table 2 Number of euchromatic SVs in the sequenced DSPR founder strains and Oregon-R**

| Strains | TE | Duplication CNV | Indels | Inversion |
|---|---|---|---|---|
| A1 | 620 | 144 | 584 | 4 |
| A2 | 618 | 123 | 785 | 10 |
| A3 | 580 | 134 | 702 | 11 |
| A4 | 581 | 136 | 683 | 7 |
| A5 | 597 | 122 | 700 | 10 |
| A6 | 760 | 136 | 681 | 10 |
| A7 | 629 | 184 | 916 | 8 |
| AB8 | 606 | 121 | 660 | 10 |
| B1 | 646 | 129 | 699 | 6 |
| B2 | 687 | 132 | 633 | 7 |
| B3 | 624 | 147 | 720 | 8 |
| B4 | 656 | 120 | 646 | 4 |
| B6 | 683 | 116 | 682 | 4 |
| Ore | 518 | 135 | 621 | 14 |

**SVs are common in genes and enriched at mapped QTLs.** In order to illustrate how common SV genotypes are in heterozygous individuals, we quantified the per gene SV burden per synthetic diploid *D. melanogaster* individual (Fig. 2a, b; each

synthetic diploid is one of 78 possible pairings of the thirteen assembled DSPR founders). On average, SVs appear in 9.3% of genes in diploid individuals (1285/13761). Of those, more than a third of burdened genes in diploids (443/1285) bear multiple SV mutations. One or more SVs burden more than half of genes in and above the 20-35kb range (Fig. 2a). Furthermore, individual genes bearing multiple SVs comprise more than a third of burdened genes between 20 and 35 kb in length and more than half

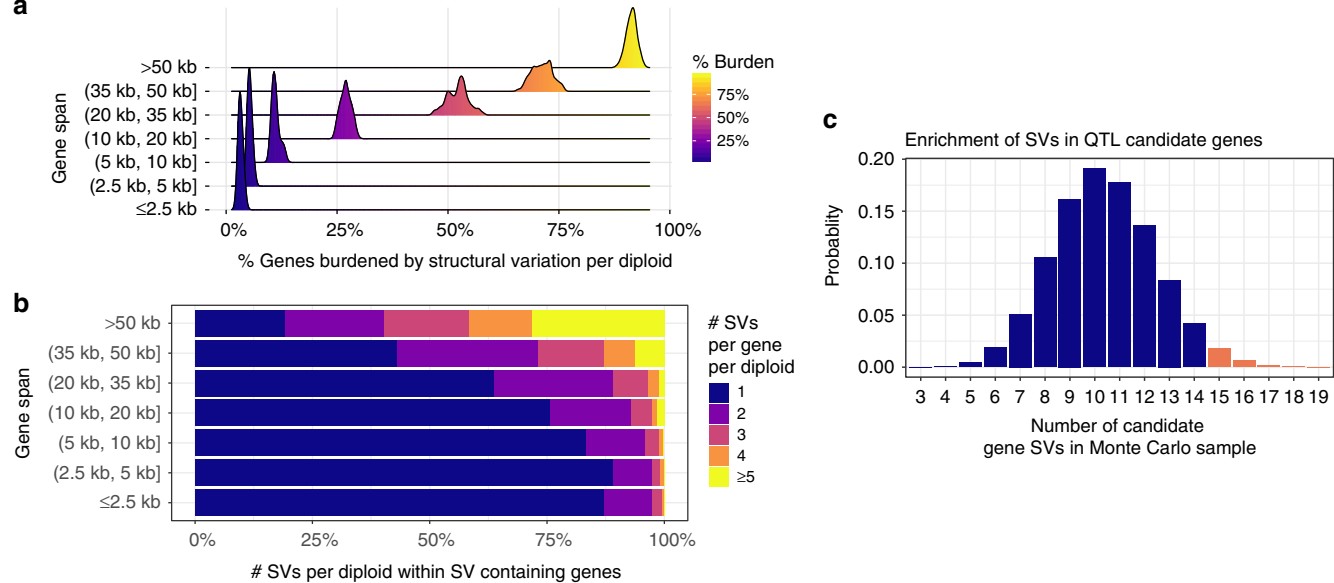

**Fig. 2** Ubiquity of SVs in *D. melanogaster* genes and QTL candidate genes. **a** Structural variant burden in diploids. The *x*-axis describes the percentage of genes in a particular length category that carry one or more SVs in diploid individuals. The distributions are derived from the collection of all 78 possible diploids that can be constructed from crosses of the 13 founders reported here. Number of genes in each length bin is in supplementary Data 4. **b** Structural variant multiplicity in diploids. The *x*-axis describes the number of variants per diploid individual in SV carrying genes. The *y*-axis describes the length of the gene span. **c** Monte Carlo distribution of number of QTL candidate genes possessing at least one SV in a sample of 31 genes. In total, 2.6% of the samples exhibit at least 15 genes harboring SVs (red bars). Genes are randomly chosen such that the gene length distributions of the Monte Carlo samples are the same as the observed candidate genes. In the empirical dataset, 15 QTL candidate genes possessed one or more SVs

of larger genes (Fig. 2b). Thus, although generally having rare minor allele frequencies, SVs are ubiquitous in the functional elements of *D. melanogaster* genome.

Although hypotheses employing SVs to explain missing heritability have been proposed[10,46], the systematic under-identification of SVs via short read- and microarray-based genotyping[21] limits their explanatory power. Using our comprehensive SV map, we measured the prevalence of SVs at the candidate genes reported in eight complex trait mapping experiments employing DSPR (Supplementary Data 1). We consider only genes in mapped QTLs explicitly cited by the authors of the original QTL studies (Supplementary Data 1; see Methods). In total, we identified 31 candidate genes of which 15 (48.4%) possess at least one SV in one founder strain, whereas only 23.4% (3237/13,830) of other *D. melanogaster* genes harbor SVs ($p = 0.0023$; Fisher's exact test). The 31 candidate genes from QTL mapping work (Supplementary Data 1) are more than twice as large as an average *Drosophila* gene (12.2 vs 5.4 kb, Wilcoxon rank sum test; $p = 1.4 \times 10^{-4}$. Supplementary Data 2). We then tested whether QTL candidate genes are enriched for SVs independent of gene size. To control for the observed elevated size distribution of QTL candidate genes, we randomly drew 100,000 Monte Carlo gene samples matching the candidate gene length distribution in order to generate null distribution for the number of SVs we expect to observe. In the Monte Carlo sample, 10.4 genes are expected to harbor SVs by chance whereas we observe 15, an enrichment of 45% ($p = 0.026$; Fig. 2c). Similarly, enrichment is also observed when burden is measured as density (# burdened genes/bp) in comparison of QTL candidate genes to the remaining genes (39.6 burdened genes/Mbp of genic DNA vs 27.3 burdened genes/Mbp of genic DNA, $p = 0.017$; Supplementary Data 2). Finally, candidate genes also exhibit greater SV mutation density compared to the expected SV density in the Monte Carlo samples (205.8 SVs/Mbp vs 138.4 SV/Mbp, $p = 0.047$; Supplementary Data 2), an enrichment of 49%. These observations suggest that SVs may be disproportionately associated with QTL candidate genes even after accounting for gene size. However, due to the limited sample size of QTL candidate genes, the power to detect even strong effects (e.g. the 45% enrichment reported here) is limited. Consequently, further QTL studies identifying candidate genes would substantially improve our understanding of this effect. These observations suggest that the contribution of rare SVs of large effect to complex traits could be pervasive.

**Functional structural variation at mapped QTL.** GWAS experiments are poorly powered to detect the segregation of multiple alleles at a causal gene[43]. Although allelic heterogeneity can be readily identified in multi-parent panels (MPPs) via QTL mapping[25], mapping resolution is often poor, forcing investigators to identify mutations of obvious functional significance in the genomic interval most likely to harbor the QTL. Both GWAS and QTL mapping suffer if putatively causative SVs disproportionately escape detection by short read sequencing[15]. This limitation can be readily solved in MPPs, as the SV genotypes of the large panel of mapping lines can be imputed from de novo assemblies of the much smaller number of MPP founders[24].

A nicotine resistance mapping study employing the DSPR identified differentially expressed cytochrome P450 genes *Cyp28d1* and *Cyp28d2* as candidate causative genes at a mapped QTL, but proposed no causative mutations[47]. A previous de novo assembly of single DSPR founder strain identified a resistant allele possessing tandem copies of the *Cyp28d1* gene separated by an Accord LTR retrotransposon fragment[15] (Fig. 3a; Supplementary Fig. 11). Our assemblies of additional DSPR founder strains reveal a total of seven structurally distinct alleles in this region, including additional candidate resistant alleles harboring gene duplications (Fig. 3a–b). For example, the resistant strain A2 carries a tandem duplication of a 15Kb segment containing both

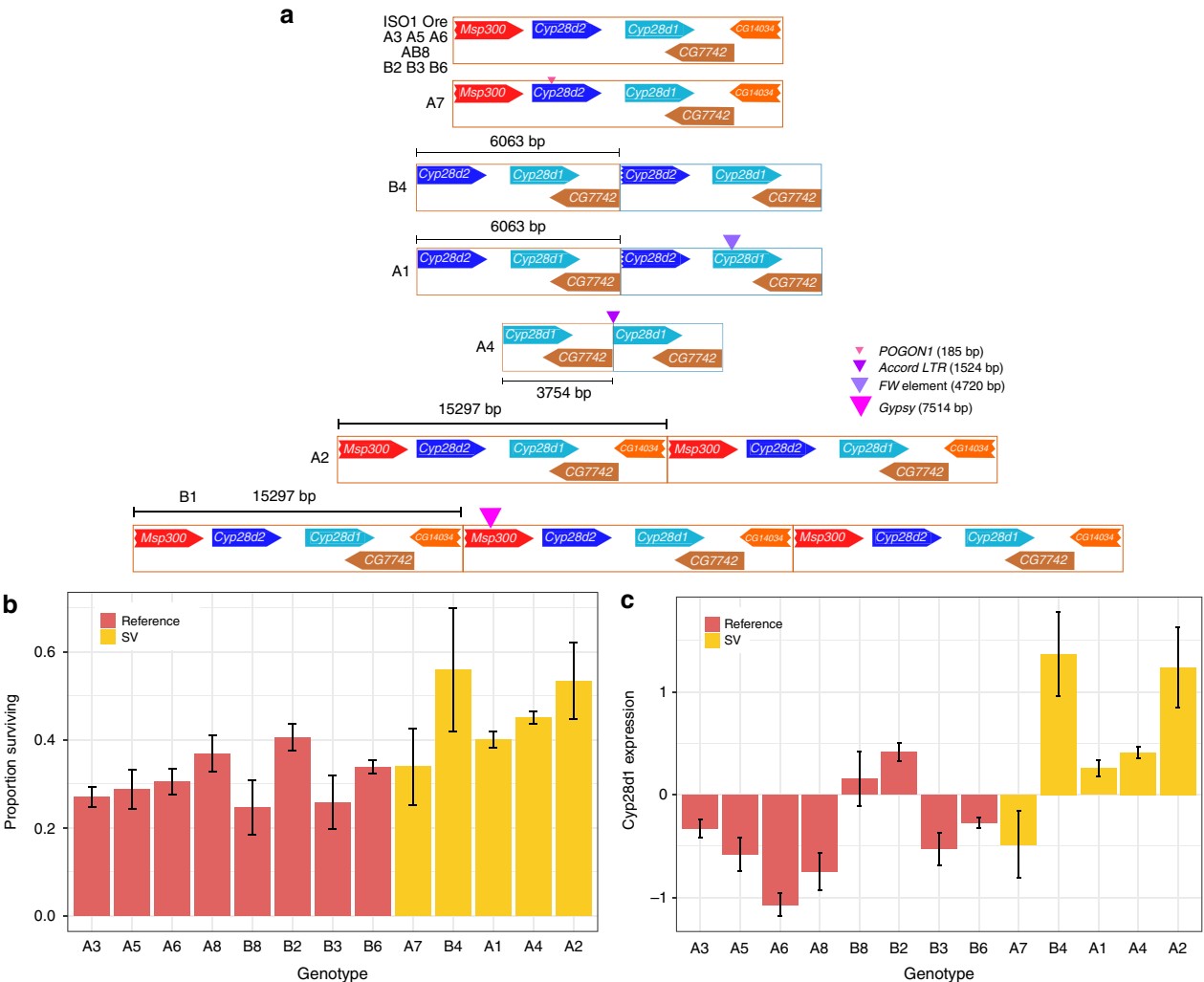

**Fig. 3** SV allelic heterogeneity at the nicotine resistance QTL candidate genes *Cyp28d1*& *Cyp28d2*. **a** Structural alleles at *Cyp28d1* and *Cyp28d2* genes underlying a nicotine resistance QTL. The most common allele is the reference (ISO1) allele and contains a single copy of *Cyp28d1* and *Cyp28d2*. All others are private to a single founder strain. A1 and B4 carry the same duplicate but A1 has a FW element inserted into the second *Cyp28d1* copy. A2 and B1 possess two and three copies of a 15 Kb segment,respectively, that contains both *Cyp28d1* and *Cyp28d2*. **b** Nicotine resistance of RILs as a function of founder allele at the QTL harboring *Cyp28d1* and *Cyp28d2*. Genotype ordering matches Fig. 2a. A2 and B4 alleles are most resistant to nicotine, followed by A4. A2 possesses a 15 Kb duplication containing full *Cyp28d1* and *Cyp28d2*, whereas B4 contains duplicate of a full *Cyp28d1* and near complete *Cyp28d2*. No RIL homozygous for the B1 allele was present in this sample. **c** Normalized *Cyp28d1* expression level in RILs with different founder genotypes at the cis-eQTL for *Cyp28d1* (Supplementary Fig. 12). Genotype ordering (L to R) follows genotype ordering (top to bottom) in Fig. 2a. A2 and B4 genotypes, which show highest resistance to nicotine toxicity, also show highest upregulation for *Cyp28d1*. Despite A1 and B4 having the same duplication, the *Cyp28d1* disrupting TE insertion in A1 is likely responsible for lower expression of the gene in A1. Error bars indicate standard errors of genotypic means

*Cyp28d* genes. The expression level of *Cyp28d1* in the adult female heads of RILs bearing the A2 genotype is highest among all founder genotypes measured (Fig. 3c). Consistent with this, DSPR Recombinant Inbred Lines (RILs) bearing the A2 genotype show the highest average resistance to nicotine toxicity among the RILs derived from the A set of founders[47] (Fig. 3b). This implies that the extra copies of *Cyp28d1* and/or *Cyp28d2* account for the increased expression and concomitant resistance to nicotine. Similarly, the B4 allele comprises a tandem duplication of a 6 Kb segment, containing one extra copy of *Cyp28d1* and a nearly complete copy of *Cyp28d2* (Fig. 3a; Supplementary Fig. 11). RILs carrying the B4 genotype at the *Cyp28d* locus also show high resistance to nicotine, making the duplication a compelling candidate for the causative mutation. On the other hand, in two alleles, TE insertions disrupt *Cyp28d* gene structure. For instance, A1 has a duplication sharing the same breakpoints as B4, but a 4.7Kb F element inserted in the 5th exon disrupts the protein

coding sequence of the second *Cyp28d1* copy, likely rendering the copy nonfunctional (Fig. 3a; Supplementary Fig. 11). Consistent with the hypothesis that the duplication causes increased nicotine resistance, the A1 genotype is more susceptible to nicotine than B4 (Fig. 3b). All of these SV alleles are singletons, and thus represent a hidden allelic series composed of individually rare alleles.

SVs may also affect genes central to life history traits. Expression levels of the insulin signaling pathway genes show substantial variation in F1 hybrids between DSPR panel B RILs and the A4 founder[48]. Among these is *Insulin Receptor* (*InR*), which plays a key role in several life history traits related to lifespan and is likely a key molecular mediator of the tradeoff between reproductive success and longevity[49–51]. Amino acid polymorphism in *InR* evolves under positive selection and some non-synonymous variants affect fecundity and stress response[51,52]. Expression variation of *InR* also affects body size,

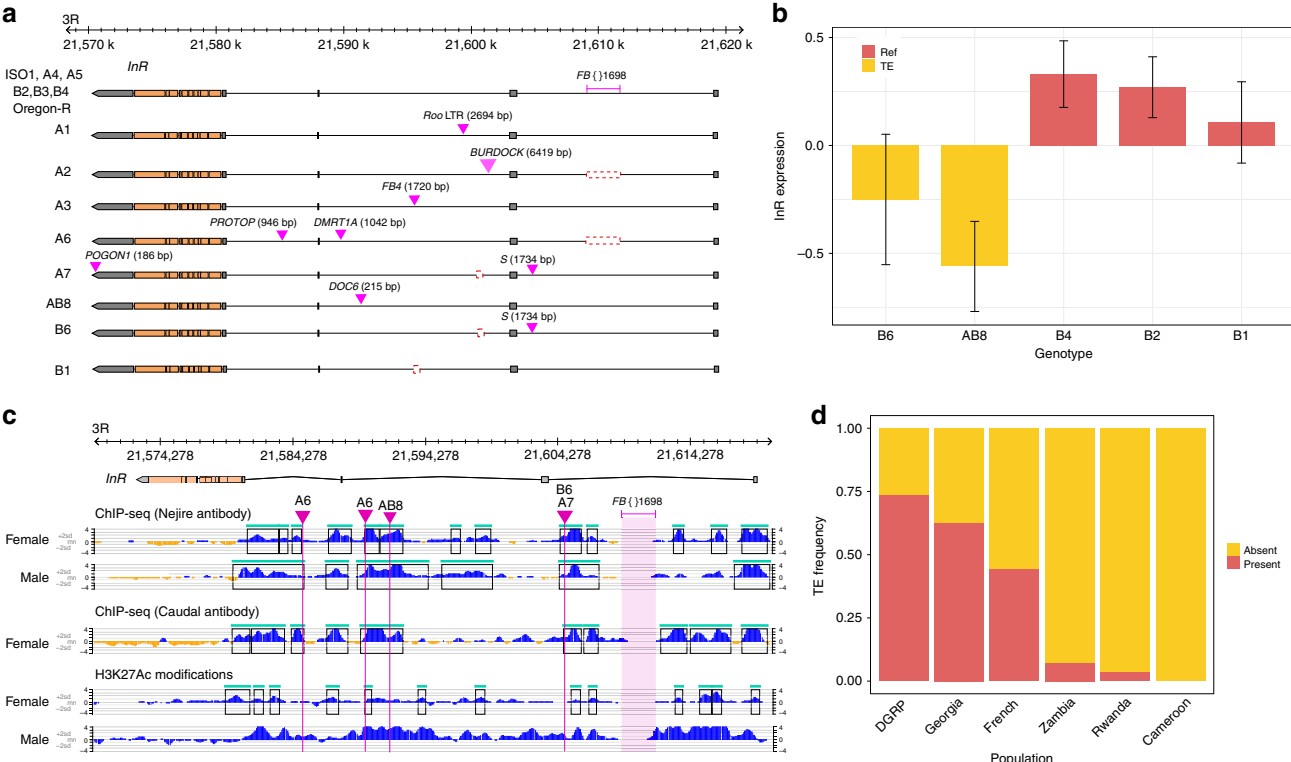

**Fig. 4** SV allelic heterogeneity at *InR* and their functional consequences. **a** Eight structurally different alleles consisting of different intronic TEs at Insulin receptor (*InR*) gene. The reference TE FB{}1698 is present in all sequenced strains, except A2 and A6. In contrast, most of the other TEs are private to the founder strains carrying them. The S element in A7 and B6 and the Doc6 element in AB8 insert into known cis-regulatory sequence. **b** Expression level of InR in F1 hybrids between panel B RILs and the founder strain A4. RILs with AB8 and B6 genotype show downregulation of InR among the panel B genotypes. Both AB8 and B6 harbors TEs that insert into known cis-regulatory intronic sequence of InR. **c** Insertion of TE insertions into transcription factor binding sites (TFBS) for Caudal and Nejire. The top and middle panel shows the TF binding peaks detected from ChIP-seq performed with Nejire and Caudal antibodies, respectively. The bottom panel shows H3K27Ac histone modifications representing a transcriptionally active state. The histone marks largely overlap with the TFBS, supporting the functional significance of the latter. High frequency FB{}1698 is inserted between two TFBS enriched sites. Disruption of TFBS by transposon insertion in B6 and AB8 InR alleles likely cause downregulation of the gene as shown in 3b. **d** Frequency of FB { } 1698 insertion in InR in different cosmopolitan and ancestral populations of D. melanogaster. The TE insertion is rare in the ancestral African populations but segregates in intermediate to high frequencies in the derived, cosmopolitan populations. Error bars indicate standard errors of genotypic mean

lifespan, and fecundity[53,54], suggesting that natural cis-regulatory variation might also be under selection. We discovered a 215 bp fragment of a DOC6 element within a second intron enhancer[55] (Fig. 4b, c) of *InR* on the AB8 haplotype, and this allele exhibits reduced gene expression relative to reference genotypes (Fig. 4b). This mutation potentially disrupts the enhancer (Supplementary Fig. 12), making it a plausible candidate for expression variation in *InR*. Another founder, A6, carries a 1,042 bp insertion of DMRT1A (LINE) in the 2nd intron and a 946 bp insertion of a fragment of PROTOP in the 3rd intron. Both insert within known cis-regulatory elements[55] (Fig. 4b, c). Except for A2 and A6, all strains, including ISO1, harbor an FB-NOF element (FB{}1698) inside the first intron of *InR* (Fig. 4a). Like many genes, the first intron of *InR* possess several transcription factor binding sites (TFBS), including those for factors *Nejire* and *Caudal*[56] (Fig. 4c). The FB-NOF element is inserted within this dense cluster of TFBS and active enhancer marks (Fig.4c). Furthermore, the FB element is segregating at high frequency in the strains discussed here (13/15), a North American population[57] (125/170), and a French population[58] (4/9), but is rare in populations derived from *D. melanogaster's* ancestral range in Africa[58,59] (Cameroon: 0/10, Rwanda: 1/27, Zambia: 10/139) (Fig. 4d). This raises the possibility that the FB element is more common in temperate cosmopolitan populations, similar to a previously described adaptive amino acid variant in *InR*[52]. In total, *InR* harbors a

remarkable amount of potentially functional structural diversity. Including these variants described above, there are nine TE insertions and two deletions throughout the gene, many of which impinge on candidate regulatory regions or transcribed portions of the gene (Fig. 4a, c).

Public resources like modENCODE annotate molecular phenotypes (e.g., RNAseq, ChIPseq, DNase-seq) against reference genomes which are often genetically different than the strains assayed[26–28,56]. Canton-S (our DSPR founder A1) and Oregon-R are strains commonly used in phenotypic assays[26–28], and we observe SVs segregating between these two strains and the reference (Table 2). Interpretation of functional genomics data such as RNA-seq can be misleading when gene copy number varies between strains. We explored the glutathione synthetase region (containing *Gss1* and *Gss2*), which is just one example among hundreds in modENCODE that likely suffer from misleading annotations. A tandem duplication present in ISO1 has created two copies of *Gss1* and *Gss2*, which are associated with toxin metabolism and linked to tolerance to arsenic[60] and ethanol induced oxidative stress[61]. While this duplication segregates at high frequency in DSPR strains (9/13), it is absent in Oregon-R (Fig. 5a) and escapes detection via short-read methods. As a result, using transcript and ChIP data derived from Oregon-R (as used in modENCODE[27,28]) results in misleading annotations of the two copies in ISO1. Indeed, among the eight

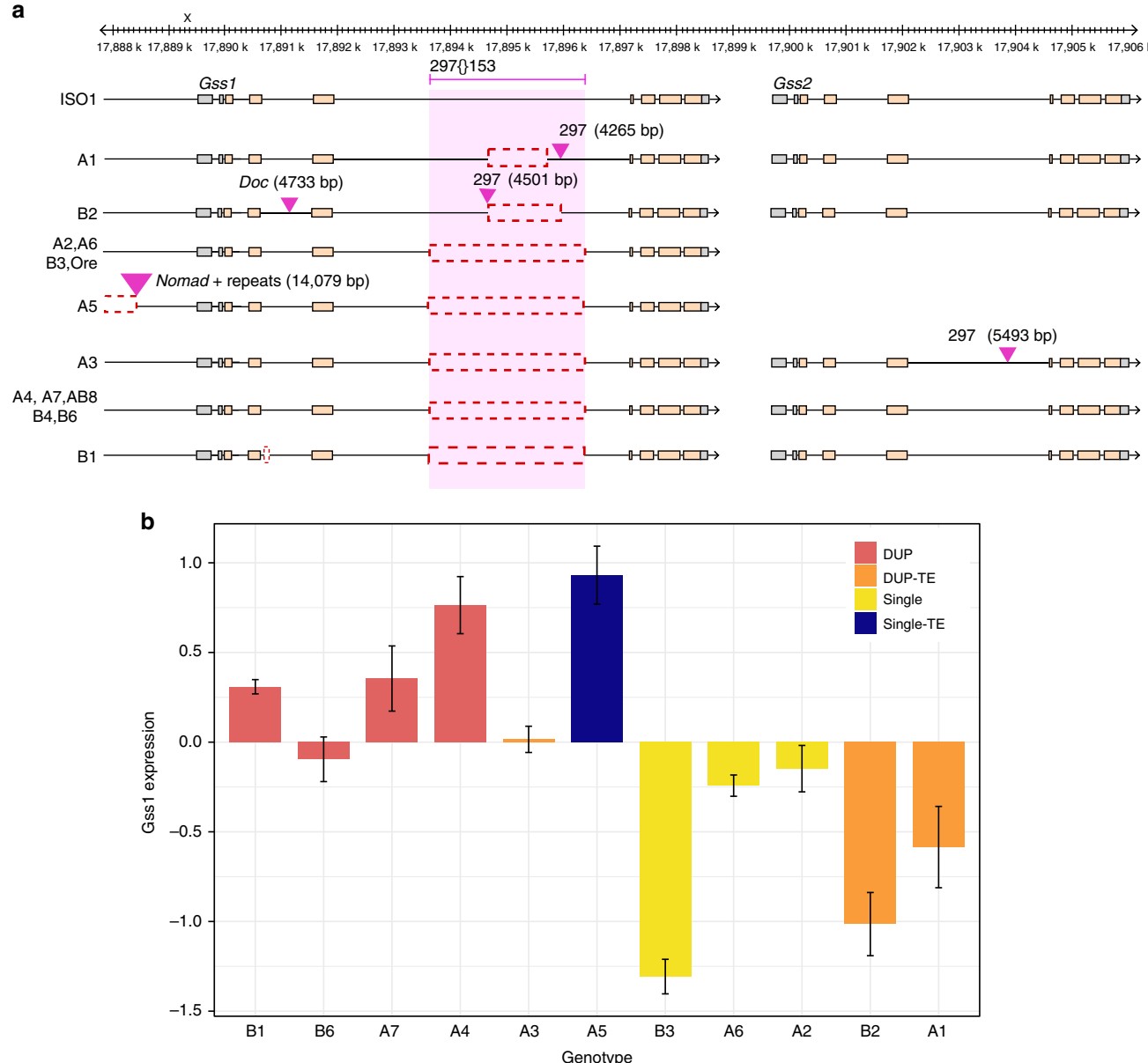

**Fig. 5** SV allelic heterogeneity at a arsenic toxicity resistance candidate *Gss1* and *Gss2*. **a** Structurally distinct alleles at Glutathione synthetase (*Gss*) locus. Nine founder alleles consist of duplication of *Gss*, among which four also carries insertion of the LTR retrotransposon 297. Among the five single *Gss* copy alleles, A5 possess a 14 Kb insertion comprising Nomad LTR TE fragments and simple repeats located 1064 bp 5′ to the transcription start site of Gss1. A5 also carries a deletion that removes 3489 bp (X:17,884,993-17,888,482) from the 5′ upstream region of *Gss1*. **b** Normalized expression level of *Gss1* for different founder genotypes. All founder alleles, except A5, possessing single copy *Gss* show varying levels of downregulated transcript levels. We hypothesize that the indel upstream of the *Gss1* transcription start site (TSS) in A5 up-regulates expression. Error bars represent standard errors of the genotypic means

structurally distinct *Gss* alleles in our dataset, ISO1 is the sole representative of its allele (Fig. 5a). The two most common *Gss* alleles include one that contains only a single *Gss* gene (in four strains, including Oregon-R) and one carrying only a tandem duplication, creating the *Gss1/Gss2* pair (in five strains, including Samarkand/AB8) (Fig. 5a). The remaining six alleles have SV genotypes represented by only a single individual in the sample. Collectively, this sample represents a haplotype network of structural variation involving five TE insertions, one duplication, one insertion comprising TE and simple repeats, and two non-TE indels. The single copy allele with a 5′ insertion of a 14 kb repetitive sequence comprising *Nomad* retrotransposon fragments exhibits the highest expression, followed by duplicate

alleles, whereas single copy alleles and duplicate alleles with intronic TE insertions generally have the lowest expression levels (Fig. 5b).

## Discussion

Despite claims that a significant proportion of complex trait variation in humans, model organisms, and agriculturally important animals and plants is likely due to rare SVs of large effect[11], systematic inquiry of this hypothesis has been impeded by genotyping approaches attuned to SNP detection[21]. As reference quality de novo assemblies of population samples for eukaryotic model systems become increasingly cost-effective,

methodical evaluation of the contribution of SVs to the genetic architecture of complex traits becomes feasible. Our comprehensive map of SVs in *Drosophila* provides the means to systematically quantify the contribution of rare SVs to heritable complex trait variation (Figs. 2a, 3a, 4a, and 5a). The value of comprehensive SV detection is underscored by the presence of SVs in ~50% of the candidate genes underlying mapped *Drosophila* QTL, and by the observation that a large fraction of *Drosophila* genes harbor multiple rare SV alleles. The genomes of humans and agriculturally important plants and animals harbor more SVs than *Drosophila*, and thus are likely more burdened with genic SVs.

The genetic heterogeneity hypothesis for variation in complex traits posits that a sizable fraction of human complex disease is associated with an allelic series consisting of individually rare causative mutations at several genes of large effect[62]. Furthermore, models for complex traits under either stabilizing[63,64] or purifying selection[42,43] with constant mutational input predict the existence of genes segregating several individually rare causative alleles that account for a sizable fraction of trait variation. We provide examples of SVs in genes of functional significance, and show that genes harboring SVs are overrepresented in a collection of QTL candidate genes. Hidden SVs are thus examples of collectively common but individually rare deleterious genetic variants predicted under the genetic heterogeneity hypothesis. Future de novo assemblies of other genomes, including humans, models, and agriculturally important species, would quantify the generality of observations from *Drosophila*.

## Methods

**DNA extraction**. Genomic DNA was extracted from females following the protocols described previously[22] and the genomic DNA was sheared using 10 plunges of a 21-gauge needle, followed by 10 of a 24-gauge needle (Jensen Global, Santa Barbara). All testing and research involving flies were performed in compliance with relevant ethical regulations. SMRTbell template library was prepared following the manufacturer's guidelines and sequenced with P6-C4 chemistry in Pacific Biosciences RSII platform at University of California Irvine Genomics High Throughput Facility. The total number of SMRTcell and base pairs sequenced, and read length metrics for each strain is given in Table S6.

**Genome assembly**. The genomes were assembled following the approach described in Chakraborty et al.[22]. For all calculations of sequence coverage, a genome size of 130Mbp is assumed ($G = 130 \times 10^6$ bp). For individual strain, we generated a hybrid assembly with DBG2OLC[65] using 30X PacBio reads, and a PacBio assembly with canu v1.3[66] (Supplementary Data 3). The paired end Illumina reads were obtained from King et al.[24]. The hybrid assemblies were merged with the PacBio only assemblies with *quickmerge* v0.2[22,67] ($l = 2$ Mb, ml = 20000, hco = 5.0, c = 1.5), with the hybrid assembly being used as the query. Because the PacBio assembly sizes were closer to the genome size of *D. melanogaster*, we added the contigs that were present only in the PacBio only assembly but not the hybrid assembly by performing a second round of *quickmerge*[67]. For the second round of *quickmerge* (l = 5 mb, ml = 20000, hco = 5.0, c = 1.5), the PacBio assembly was used as the query and the merged assembly from the first merging round the reference assembly. The resulting merged assembly was processed with *finisherSC* to remove the redundant sequences and additional gap filling using raw reads[68]. The assemblies were then polished twice with *quiver* (SMRTanalysis v2.3.0p5) and once with *Pilon v1.16*[69] using the same Illumina reads as used for the hybrid assemblies.

**Comparative scaffolding**. We scaffolded the contigs for each assembly based on the scaffolds from the reference assembly[70], following a previously described approach[15]. Briefly, TEs and repeats in the assemblies were masked using RepeatMasker (v4.0.7) and aligned to the repeat-masked chromosome arms (X, 2L, 2R, 3L, 3R, and 4) of the *D. melanogaster* ISO1 assembly using MUMmer[71]. After filtering of the alignments due to the repeats (delta-filter −1), contigs were assigned to specific chromosome arms on the basis of the mutually best alignment. The scaffolded contigs were joined by 100 Ns, a convention representing assembly gaps. The unscaffolded sequences were named with a "U" prefix.

**BUSCO analysis**. We ran BUSCO (v3.02)[72] on the Pilon polished pre-scaffolding assemblies to evaluate the completeness of all the assemblies relative

to the ISO1 release 6 (r6.13) assembly. We used both the arthropoda and diptera datasets for the BUSCO evaluation. For the arthropoda database, three orthologs (EOG090X0BNZ, EOG090X0M0J, and EOG090X049L) were not found in any of the 15 strains (ISO1, Oregon-R, and 13 DSPR founders). Further inspection of these orthologs revealed that they are present in ISO1 even though the BUSCO analysis misses them when applied to ISO1 (EOG090X0BNZ is CG3223, EOG090X0M0J is Pa1 and EOG090X049L is CG40178). Consequently, we removed these three genes from consideration as uninformative.

**Variant detection**. For variant detection, we aligned each DSPR assembly individually to the ISO1 release 6 assembly (release 6.13)[70] using nucmer (nucmer –maxmatch –noextend)[71]. We identified and classified the variants using SVMU 0.2beta (Structural variants from MUMmer) ($n = 10$)[15]. SVMU classifies the structural differences between two assemblies as insertion, deletion, duplication, and inversion based on whether the DSPR assemblies have longer, shorter, more copy, or inverted sequence, respectively, with respect to the reference genome. The variant calls for individual genomes were combined using bedtools merge[73] and converted into a vcf file using a custom script (https://github.com/mahulchak/dspr-asm). TE insertions were identified by examining the overlap between RepeatMasker identified TEs and SVMU insertion calls using bedtools, requiring that at least 90% of RepeatMasker TE annotation overlap with svmu insertion annotation. 12.8% SV mutations, for which mutation annotation were complicated by secondary mutations, were flagged as "complex" (CE = 2 in the VCF file). Additionally, 16.3% SVs that were located within 5Kb of a complex SV were often part of a complex event and were also assigned a tag (CE = 1) to differentiate them from the unambiguously annotated SVs (CE = 0).

**Genotype validation**. To determine the genotyping error rate, a set of randomly selected 50 simple (CE = 0) SVs obtained from SVMU were manually inspected on UCSC genome browser representation of the multiple genome alignment of the 15 genomes (http://goo.gl/LLpoNH). Furthermore, to estimate the genotyping accuracy of the SVs occurring in the vicinity of the complex mutations, where mutation annotation is complicated by alignment ambiguities, we manually inspected 217 SVs occurring within 20 Kb of 50 randomly selected complex (CE = 2) SVs. Among these, 3/217 and 0/50 SVs were absent in the UCSC browser and therefore they are likely mis-annotated by our pipeline. The mis-annotated SVs (insertion in A1 and tandem array CNV in A7) are located in a complex, repetitive, structurally variable genomic region on chromosome 3L (3L:7669500-7679100) (Supplementary Fig. 5).

**Comparing SV genotypes from de novo assemblies to short read only calls**. TE genotypes for the founders[18] were downloaded from flyrils.org and the insertion coordinates were lifted over to the current release (release 6) of the reference genome[70] using UCSC liftover tool[74]. For detection of the duplicates, we have previously found that discordant read pair based method (Pecnv)[75] was comparable to split read mapping[76] and more reliable than methods based on coverage alone[15,77], so we used Pecnv. Pecnv was run using the settings described before[15]. Because svmu reports tandem duplicate CNVs as insertions (with appropriate CNV tags to separate from TE and other insertions) and Pecnv reports sequence range being duplicated, the SVMU CNV insertion coordinates were extended by 100 bp before comparison (bedtools intersect) between Pecnv output and svmu output was conducted. The non-TE indel genotypes were obtained from Pindel output (the "LI" and "D" events) using the commands described previously[15]. For determining population frequency of indel SVs (e.g. the reference FB element in *InR*), Pindel output based on the alignment bam files were used. We only estimate the false negative rate of short read only callers, but note that these methods also generate false positive SV calls.

**Gene expression analysis**. The preprocessed expression data for female heads[78] and IIS/TOR expression data[48] from whole bodies were downloaded from www.flyrils.org. Expression QTL analysis (Supplementary Fig. 11) for *Cyp28d1* and *Gss1* using the head expression data were performed using the R package DSPRqtl following the instructions provided in the manual (DSPRscan,model = gene ~ 1,design = "ABcross"). When expression data for multiple isoforms were present, expression data only for the longest transcript that is expressed in the head was used. The genotype values at the eQTL were determined using the function DSPRpeaks included in the DSPRqtl package. No eQTL were found for *InR* so the genotype values for the *InR* expression data were obtained by assigning the founder genotypes to the RILs used in the IIS/TOR expression dataset, using the posterior probabilities of the forward-backward decoding of the HMM for the panel B RILs available on www.flyrils.org. *Drsl5* expression levels in A4 and A3 were obtained from a publicly available RNAseq dataset[47].

**Comparison of site frequency spectra**. The histogram of allele frequencies (site frequency spectrum or SFS) was collated for four categories: synonymous SNPs, non-synonymous SNPs, duplicate CNVs, and TE insertions. The frequencies of SNPs were collected from the VCF file[24] using vcftools and bcftools[79,80]. The frequencies of SVs were collected from the column 4 of the combined SVMU

output for the TE insertions and duplication CNVs from all DSPR strains (https://github.com/mahulchak/dspr-asm). Complex mutations (CE = 1 and CE = 2) were excluded from the analysis. Let $N$ be the sample size and $x_i$ be the number of sites in frequency class $i$, where $0 < i < N$. The SFS was "folded", meaning we focused attention on the minor allele frequency (MAF), or $y_i$ = minimum $(x_i, N - x_i)$. Pairwise comparisons between different SFS site categories were conducted using the $\chi^2$ test on allele frequencies and site categories. For allele frequencies, two types of classifications were used: (1) every $y_i$ for $0 < i < N$ ($N - 1$ df); and (2) considering singletons versus the other frequency categories, or $y_i$ for $i = 1$ versus $2 < i < N$ (1 df).

**Candidate genes associated with mapped QTL.** The candidate genes from DSPR QTL papers were selected based the following criteria: (1) The gene falls within the QTL peak; (2) additional functional data is cited by the authors of the respective study to highlight the gene; (3) the functional information cited by the authors did not use knowledge about structural variation affecting the candidate locus (Supplementary Data 1). The additional data can either be expression data collected by the authors or existing functional data known about the genes. Only 44 candidate genes from eight studies fulfilled these criteria but three among them fell outside the euchromatic boundaries used here (Supplementary Table 1). Hence only 41 candidate genes were included in the SV enrichment analysis. Of the 41 candidate genes identified, 10 of them were at a single locus (GstE1-10). As a result, we carry out our analysis treating GstE1-10 as either a single gene or ten different genes (the qualitative outcome is unchanged). To test if candidate genes are longer than average genes, we considered all genes (Supplementary Data 2) as well as the dataset excluding the GstE1-10 genes (Supplementary Data 2). The lengths of candidate genes were compared against the rest of the genome using a Mann–Whitney U test.

**Candidate gene enrichment analysis.** To determine if candidate genes are enriched for SVs relative to the rest of the genome, we analyzed the candidate gene dataset without the GstE1-10 gene array (Table S4). The genes comprising GstE1-10 were excluded to avoid confounding effects of the complex structure of the locus and the identity of GstE1, GstE3, GstE5, and GstE6. As described below, the conclusions of the enrichment analysis does not depend on this locus, so we report the results of excluding it. A Fisher's Exact Test was applied to the counts in categories of candidate gene vs. rest of the genome and SV-free vs. SV-burdened genes. To account for the lengths of the candidate genes being longer than the rest of the genome, we performed a Monte Carlo sampling of the whole genome according to the histogram of gene sizes in the candidate gene list (Supplementary Data 2). We sampled from the genome by drawing from each gene length bin according to a hypergeometric distribution, where n is the number of candidate genes in the candidate bin, K is the number of SV-burdened genes in the genome bin, and N-K is the number of SV-free genes in the genome bin (Supplementary Data 2). We then tallied up the number of observed SVs across all bins. We repeated this 100,000 times to construct a Monte Carlo distribution of the SV burden expected of genes matching the size distribution observed in the actual candidate genes. This led to 100,000 simulated size distributions that matched the observed size distributions (every Mann–Whitney U $p$-value of Monte Carlo sample lengths compared against the observed candidate lengths >0.1). Expected density of SV-burdened genes (number of burdened genes per Mbp of gene spans) and expected SV density (total number of SVs per Mbp of gene spans) were also calculated from the Monte Carlo samples. Although we present the enrichment results from analysis performed on the gene set without the GstE1-10 array, inclusion of the array either as a single 13kb locus or as individual genes does not alter the conclusion of the enrichment analysis (single Gst locus: enrichment $p$-value = 0.021, length $p$-value = $6.5 \times 10^{-5}$; individual Gst genes: length $p$-value = 0.034 and enrichment $p$-value = $2.9 \times 10^{-3}$).

**Calculating the SV burden in genes in diploid individuals.** In order to calculate the distribution of SV burden expected in diploids, the haploid genotypes of each founder was paired with every other founder, for a total of 78 possible pairings. For each of these diploid pairings, the number of unique SV mutations for each gene in the genome was recorded. A mutation is said to affect a gene if it falls within the gene span, which is defined as affecting nucleotides between the start and end coordinates of the gene feature in the *D. melanogaster* release 6.16 gff file[36]. The number of SV mutations overlapping a gene in a given diploid combination is considered that gene's multiplicity for that combination. Any gene with a multiplicity ≥1 for a particular diploid comparison is considered SV-burdened for that diploid.

**Reporting summary.** Further information on research design is available in the Nature Research Reporting Summary linked to this article.

## Data availability
All scaffolded assemblies and the raw data (HDF5 files and their respective metadata) have been deposited in NCBI under the Bioproject accession PRJNA418342. All raw SV outputs, and processed data are available at https://github.com/mahulchak/dspr-asm.

## Code availability
All scripts and codes have been deposited to GitHub and freely accessible from https://github.com/mahulchak/dspr-asm.

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

## Acknowledgements

We wish to acknowledge support from the following grants OD010974 (SJM and ADL), GM115562 (ADL), R01GM123303-1 and University of California, Irvine setup funds (JJE), and K99GM129411 (MC). We thank Luna Thanh Ngo and Daniel Na for help with data management and fly maintenance. This work was made possible, in part, through access to the Genomics High-Throughput Facility Shared Resource of the Cancer Center Support Grant CA-62203 at the University of California, Irvine, and NIH shared-instrumentation grants 1S10RR025496-01, 1S10OD010794-01, and 1S10OD021718-01.

## Author contributions

M.C., J.J.E., S.J.M., A.D.L. conceived of the work and wrote the paper. M.C. assembled the genomes and wrote the variant caller.

## Competing interests

The authors declare no competing interests.
