## [Peer Review File · Nature Communications]

Reviewers' comments:

Reviewer #1 (Remarks to the Author):

This is a fairly simple, descriptive paper, characterizing the genome sequences of some 14 lines of *Drosophila melanogaster*. This may not seem new or exciting, but there are many important messages here, and those messages come from a huge amount of careful analysis. The sequences are each de novo assemblies of deep PacBio reads, so they are of exceptional quality. Because they are long reads, often entire chromosome arms are contained in single contigs. This makes the data exceptionally good for inference of Structural Variants, including transposable element insertions. All told, the authors catalog more than 20,000 structural variants, and the list of their attributes, allele frequencies, etc., present a compelling picture for the importance of this class of mutation. While we may think that we knew most of these lessons, this paper delivers the hard numbers in impressive fashion.

Inference of the deleterious consequences of SVs is made by examining the site frequency spectrum. Because SVs larger than 5 kb are heavily skewed toward singletons, the argument is they are on average deleterious, and this drives them to low frequency. In fact, SVs of this size tend to be rarer than amino acid replacements, which is widely considered to be due to their (generally) deleterious effects.

The demonstration that some regions of the genome known to be under selection harbor a rich allelic series with multiple SVs was especially thought-provoking. Genes like the insulin receptor (InR), *Gss1/Gss2*, and *Cyp28d1* and *Cyp28d2* appear to have multiple TE insertions that disrupt expression, consistent with independent origin of many adaptive variants. The relatively high frequency of TE movement in flies makes them particularly prone to being the causal factor giving rise to these adaptive mutations.

I have just two comments for consideration for improvement. In more than one place in the manuscript there is an assumption of mutation-selection balance. I don't think there is particularly good evidence that *Drosophila* are in mutation-selection balance, mostly because of the rapid demographic shifts. It would be good to either relax this assumption, or discuss the results in light of the likelihood that mutation-selection balance might not yet be attained.

The second suggestion is that there are many other places besides the site frequency spectrum where a contrast with SNPs would be informative. How many total SNPs were called? How many were coding, how many nonsynonymous? How do the distributions of LD compare for SNP-SNP LD, vs SNP-SV, vs. SV-SV events? How many of the SVs show a signature of a selective sweep? Of the SVs that result in protein coding gene duplication, how many impact transcript abundance? It would be an interesting challenge to assess whether the variance in fitness (or in any trait) due to SNPs and that due to SVs could be assessed.

In sum, it is the thoroughness of the documentation of the magnitude and pervasiveness of structural variation in natural populations of *Drosophila* that gives this paper heft.

Reviewer #2 (Remarks to the Author):

This is a very interesting and important paper. The authors extend their prior work using PacBio to obtain reference quality de novo assembly of a *Drosophila melanogaster* strain to identify structural variants. These SVs are invisible to short read sequencing technologies. The authors now sequence and assemble 14 genomes and by pairwise whole genome alignment, they identify thousands of polymorphic SVs. Analyses of these SVs reveal that 1) they are more deleterious based on SFS; 2) they overlap with QTL genes; several examples are given including *Cyp28d*, a nicotine resistance gene; InR; Glutathione synthetases. The examples are particularly interesting and make lots of sense. There are often multiple alleles at the same locus. They provide strong evidence that SVs are underlying QTLs in many cases. Although it could be criticized as cherry-picked examples, I believe that these aren't the only cases where SVs have phenotypic consequences. The paper is extremely well written and easy to follow. The figures are of excellent

quality and greatly facilitate understandings. I therefore recommend acceptance pending some minor revisions.

The main problem I have is with the global analysis (first two figures) and some of the general claims, which I detail below.

1) The 14 genomes are from several continents so they likely represent populations that have diverged, so much so that the SFS isn't really a good measure of the selection mutation balance, but rather population differentiation. So more diverged populations will have more skewed distribution. Please note that I don't doubt the strong fitness effects of SVs, it's just that this analysis isn't particularly compelling. In addition, I don't know how SNPs are called among these assemblies, my impression is that SVs and SNPs are called under completely different models. For example, some SNP call models favor SNPs that have more common alleles and the genome-by-genome SV call won't have this problem. What I'm trying to get at is that these can all explain the SFS differences, but it probably does not matter because the effect is so strong. The authors should at least acknowledge these alternative explanations.

2) The analysis and enrichment of SVs in mapped QTLs could be improved. First, I would probably provide a measure of per base SV density such that the lengths of the genes are normalized. Basically, you provide two numbers, one is the genome-wide SV density, the other is the QTL gene SV density. These numerical values will help quantify the enrichment better than p values alone. Second, I would not present the fisher's exact test without correcting for gene length difference. Third, I would perform the test without the GstE genes.

3) My third concern is the interpretation of these data. The strongest result of this paper is that SVs are enriched among mapped QTLs. At least the paper is written on the basis of the missing heritability problem and that hidden SVs could explain some missing heritability. Therefore SVs' enrichment among QTLs is a big deal. Another related question is whether they explain more than SNPs. I don't think these are well-posed questions. Of course SVs can explain some missing heritability. As individual variants, they likely have stronger effects than SNPs on average simply because they are more dramatic changes to sequences. However, given that SVs are of lower frequency, the variance due to them could still be small. These questions cannot be answered by the present study. This study does provide some useful examples, especially the presence of multiple alleles (heterogeneity) at the same QTL. Therefore I recommend that the authors add a "can" to the title before "shape the variation" and throughout the manuscript. It's less definitive but more accurate.

Reviewer #1:

*I have just two comments for consideration for improvement. In more than one place in the manuscript there is an assumption of mutation-selection balance. I don't think there is particularly good evidence that *Drosophila* are in mutation-selection balance, mostly because of the rapid demographic shifts. It would be good to either relax this assumption, or discuss the results in light of the likelihood that mutation-selection balance might not yet be attained.*

We agree that assuming mutation-selection balance in *D. melanogaster* isn't justified by the literature. Consequently, following the reviewer's suggestion, we have made revisions the relevant section of our discussion. After revision, we make it clear that rare deleterious variants can make substantial contributions to complex trait variation under a wide variety of models including those at mutation selection balance as well as those that include recent growth or bottleneck models likely relevant for both human and *Drosophila*, citing relevant literature including Simons & Sella 2014 and Lohmueller 2014^{1,2}.

The second suggestion is that there are many other places besides the site frequency spectrum where a contrast with SNPs would be informative. How many total SNPs were called? How many were coding, how many nonsynonymous? How do the distributions of LD compared for SNP-SNP LD, vs SNP-SV, vs. SV-SV events? How many of the SVs show a signature of a selective sweep? Of the SVs that result in protein coding gene duplication, how many impact transcript abundance? It would be an interesting challenge to assess whether the variance in fitness (or in any trait) due to SNPs and that due to SVs could be assessed.

In response to the reviewer's question about the SNPs, we now highlight the source of the variants³ and now mention the number of sites in the Methods section.

We share the reviewer's curiosity about the detailed patterns of polymorphism for SVs, especially as compared to SNPs (eg LD calculations and sweep inference, etc). We appreciate the suggestions and agree with the reviewer that many interesting questions remain to be explored. Indeed, we are currently pursuing many of these analyses. However, the breadth of such studies demands a series of fairly detailed analyses and additional experiments that would encompass new material at least on the order of the scope of the current manuscript, if not more. For just one example, SFS analyses are extremely sensitive to the accuracy of the site frequency spectrum, and consequently require correcting for all manner of sources of bias, such as mis-polarizing SNPs^{4,5} or accounting for ascertainment bias and error in the SFS of SVs⁶. The transcript abundance question is also an ongoing interest that will require additional experiments. Such experiments cannot simply be done on the founders, however, as this does not control for genetic background, and so the scope is substantially larger, as it must be conducted on hundreds of RILs.

Reviewer #2:

The main problem I have is with the global analysis (first two figures) and some of the general claims, which I detail below.

1) *The 14 genomes are from several continents so they likely represent populations that have diverged, so much so that the SFS isn't really a good measure of the selection mutation balance, but rather population differentiation. So more diverged populations will have more skewed distribution. Please note that I don't doubt the strong fitness effects of SVs, it's just that this analysis isn't particularly compelling.*

We agree with the reviewer that patterns of natural variation will reflect the nature of sampling, and consequently inferences about the population genetic properties of such samples should be very cautious and take such sampling into account. Consequently, we adopted an approach whereby we compare the SVs to SNPs and restrict inferences to relative statements like, "Our analysis shows that SVs are more deleterious than amino acid changing SNPs" rather than absolute statements about variants like "Our analysis shows that SVs are strongly deleterious". This takes advantage of the fact that effects introduced by shared properties of the sample like sample selection and/or demographic history won't differentially affect different variant classes.

There is a long history of statistical tests in population genetics that compare frequencies of different types of molecular events in the same set of samples in ways analogous to ours. Such analyses comparing two classes of variants within a sample are relatively robust to non-equilibrium demography⁷⁻¹⁰. Indeed, this observation has spurred the development of statistical approaches for estimation of selection parameters (e.g. (Williamson et al. 2005)¹¹). Interestingly, in Tajima's original paper⁷, the allele frequency spectrum of putative TEs was compared to that of small indels and restriction site polymorphisms in the full understanding that such comparisons are robust to violations of assumptions related demographic equilibrium. In fact, Tajima's conclusions are remarkably similar to the ones we present here, namely that TEs in *Drosophila* are more strongly deleterious than that of other mutation classes (in this case, the result extends only to the white gene). Indeed, this recognition likely inspired many of the numerous approaches that take a similar tack in comparing site classes interleaved throughout the genome (e.g. (Akashi and Schaeffer 1997; Andolfatto 2005; Williamson et al. 2005)).

In addition, I don't know how SNPs are called among these assemblies, my impression is that SVs and SNPs are called under completely different models. For example, some SNP call models favor SNPs that have more common alleles and the genome-by-genome SV call won't have this problem. What I'm trying to get at is that these can all explain the SFS differences, but it probably does not matter because the effect is so strong. The authors should at least acknowledge these alternative explanations.

We used the SNP genotypes from the original DSPR paper (King et al. 2012). Briefly, the founder lines of the original DSPR study and this work were drawn from strains decades after they were field collected, and so have experienced a great deal of passive inbreeding. Furthermore, these already inbred/low diversity strains were then passed through ~18 additional generations of strict full-sib mating³. The King paper has a figure showing residual heterozygosity in these founder

strains, which is quite low (and unpublished work shows residual heterozygosity is much lower than in DGRP¹², other widely available inbred strains, and/or inbred African strains). Consequently, SNP calls were obtained from high coverage (~50-fold coverage) short reads from each founder. This means a SNP allele private to a single founder is detected as ~50 copies of the ALT allele, whereas in all the other founders ~50 copies of the REF allele. This sort of high coverage data obtained from isogenic strains is extremely accurate. While we feel that the particular sort of bias suggested by the reviewer doesn't affect our study, we are sensitive to the larger context implied by the reviewer. Namely, genotyping biases, no matter their origin, are extremely important to acknowledge and account for. Consequently, we have revised the appropriate section to discuss this.

2) The analysis and enrichment of SVs in mapped QTLs could be improved. First, I would probably provide a measure of per base SV density such that the lengths of the genes are normalized. Basically, you provide two numbers, one is the genome-wide SV density, the other is the QTL gene SV density. These numerical values will help quantify the enrichment better than p values alone.

We share the reviewer's concerns about the gene length confounding the SV enrichment analysis. We addressed this issue in the manuscript by correcting for gene length directly in the Monte Carlo sampling, which should correct for the bias we think concerns the reviewer. Briefly, we randomly sampled 31 size-matched genes from the entire genome such that the size distributions were the same (see Methods). We then calculated the SV burden for these randomly sampled genes to generate one instance of the Monte Carlo distribution. This was repeated 100,000 times to obtain the full Monte Carlo sample. **Notably, these Monte Carlo samples exhibited a length distribution closely matching that of the QTL genes.** Individual Monte Carlo samples were not significantly different than the QTL gene length distribution using the nonparametric Mann-Whitney U test.

However, we also agree with the reviewer that providing an intuitive summary of gene and QTL densities will be more effective at communicating and quantifying the enrichment we report. Consequently, we add these numbers to the manuscript. We have now added the densities in units of number of burdened genes per megabase of gene sequence (39.6 for QTL candidates vs 27.3 for the remaining genes) and number of SVs per megabase (205.8 for QTL candidates vs 138.4 for the remaining genes) for both candidate genes associated with QTLs and length-matched control genes (Table S4 Tab "MC SV enrichment"). Monte Carlo p-values on these density estimates remain marginally significant.

Second, I would not present the fisher's exact test without correcting for gene length difference.

We understand the reviewer's concern regarding Fisher's exact test. Initially we were trying to ask whether there was evidence of any enrichment at all of SVs in QTL candidates compared to the genome, regardless of gene length. FET is an appropriate test for this hypothesis. Of course, gene length is important, so in our follow-up hypothesis, we compared QTL candidate genes with length-matched control genes from the genome and performed the Monte Carlo simulation which

corrects for size. In the manuscript, we highlight the importance of comparing QTL candidate genes with length-matched control genes, and move beyond the FET results while drawing the final conclusion of the enrichment analysis. Thus, FET provides an important initial motivation to perform a more nuanced analysis. Ultimately, however, it is of great interest to report that SVs are enriched in QTL candidate genes, even if this was entirely due to a gene target size effect. However, since our follow-up tests consistently reject the null hypothesis at a nominal p-value of 5% or less, we can't confidently conclude that this observation is driven only by gene length. We do recognize that the sample size prevents us from making this conclusion with more confidence, but these are the cards that 8 time-consuming QTL studies carried out over many years dealt us. Clearly additional QTL studies will permit a more definitive test of this hypothesis.

Third, I would perform the test without the GstE genes.

We have performed the test without the GstE genes. Our conclusions remain the same. This is because our use of the merged GstE locus in the original version resulted in an extremely long "gene" (~13.2kb), which was virtually guaranteed to receive an SV in the Monte Carlo simulations. However, in the original version, since we were conservative and treated it as having only a single mutation rather than 4 (GstE1, GstE3, GstE5, and GstE6), including it added only 1 to the enrichment. The consequence of excluding GstE1 from the simulation effectively reduced both the expected number of candidate gene SVs from the null simulation by 1 as well as the observed number. Consequently, the new p-value is 0.026 (cf 0.021). The other statistical test that relies on the status of GstE is the Mann-Whitney U test. After excluding the GstE locus, it remains significant (1.448×10^{-4}). We've updated Figure 2, both statistical tests, and the text supporting this part of the paper.

3) My third concern is the interpretation of these data. The strongest result of this paper is that SVs are enriched among mapped QTLs. At least the paper is written on the basis of the missing heritability problem and that hidden SVs could explain some missing heritability. Therefore SVs' enrichment among QTLs is a big deal. Another related question is whether they explain more than SNPs. I don't think these are well-posed questions. Of course SVs can explain some missing heritability. As individual variants, they likely have stronger effects than SNPs on average simply because they are more dramatic changes to sequences. However, given that SVs are of lower frequency, the variance due to them could still be small. These questions cannot be answered by the present study. This study does provide some useful examples, especially the presence of multiple alleles (heterogeneity) at the same QTL. Therefore I recommend that the authors add a "can" to the title before "shape the variation" and throughout the manuscript. It's less definitive but more accurate.

We agree with the reviewer concerning the caveats of our observation. The current experiment is unable to estimate the proportion of variation in complex traits due to SVs. However, in putting this finding in historical context, we should acknowledge that this hypothesis (that SVs are associated with quantitative morphological variation) has been suggested for bristle number in *Drosophila* by Mackay and Langley in 1990¹³ and replicated by Long et al. in 2000¹⁴. Both of these studies clearly predict that this is a phenomenon generalizable to more than merely bristle

number. Our analysis of 8 different QTL studies constitutes a long anticipated test of the generality of this hypothesis. Even if the result was purely driven by gene size, this would be an important contribution to this longstanding question in and of itself. That we tentatively reject the gene target size hypothesis is an interesting and perhaps unexpected result. The combination of these two observations we think justifies the fairly circumspect word “shape”.

Here is a way of restating the reviewer's concern (in perhaps a simpler way). Most SVs we detect are singletons, thus for virtually every SV, there is likely a SNP in complete LD with it. Thus in any sort of regression of genetic variants on phenotype over a set of RILs in the DSPR (or in Mackay's DGRP if SVs were identified) the two types of variants are always going to be conflated. This is a general problem with testing Rare Alleles of Large Effect models, in most systems there will often be several private alleles closely linked to one another that could explain phenotypic variation. *A lemma of this acknowledgement is that it is similarly impossible to say rare SNPs explain variation* (since they are confounded with often-times hidden SVs). Statistically this is a difficult problem and the problem likely has to be addressed experimentally (but gene replacements are difficult, and the effects of the SVs are still likely subtle on a phenotypic level, and it is unclear what phenotypes they impact).

This being said it is helpful to consider what we have accomplished. By virtue of having identified structural variants in the DSPR and representing those variants in a Santa Cruz Browser framework, investigators working on particular traits will often discover a structural variant in a candidate gene under a QTL peak. Prior to this work, those structural variants were “invisible” and investigators would focus on functionally testing SNPs and small INDELS for an effect on their phenotype of interest. Once the investigator “sees” their gene is in fact segregating a gene duplication or perhaps harboring a large TE in an intron known to contain enhancers, they may perhaps think testing the non-SV events represents a misplaced effort (although it may turn out the causative variants are indeed SNPs). So just the observation that previously invisible structural variants are segregating in the DSPR *in my gene* has a major impact on the types of follow-up work we are likely to see in the field moving forward. The ultimate resolution of the question of SNP vs SVs and standing variation will depend at least in part on detailed functional work on individual genes impacting specific phenotypes (c.f.(Schmidt et al. 2010)¹⁵) and the work of this paper is highly enabling in this regard.

References:

- 1 Simons, Y. B., Turchin, M. C., Pritchard, J. K. & Sella, G. The deleterious mutation load is insensitive to recent population history. *Nature Genetics* **46**, 220-+, doi:10.1038/ng.2896 (2014).
- 2 Lohmueller, K. E. The Impact of Population Demography and Selection on the Genetic Architecture of Complex Traits. *PLoS Genet.* **10**, doi:ARTN e1004379 10.1371/journal.pgen.1004379 (2014).
- 3 King, E. G. *et al.* Genetic dissection of a model complex trait using the Drosophila Synthetic Population Resource. *Genome research* **22**, 1558-1566 (2012).

- 4 Keightley, P. D. & Jackson, B. C. Inferring the Probability of the Derived vs. the Ancestral Allelic State at a Polymorphic Site. *Genetics* **209**, 897-906, doi:10.1534/genetics.118.301120 (2018).
- 5 Barton, H. J. & Zeng, K. New Methods for Inferring the Distribution of Fitness Effects for INDELs and SNPs. *Molecular Biology and Evolution* **35**, 1536-1546, doi:10.1093/molbey/msy054 (2018).
- 6 Emerson, J. J., Cardoso-Moreira, M., Borevitz, J. O. & Long, M. Natural selection shapes genome-wide patterns of copy-number polymorphism in *Drosophila melanogaster*. *Science* **320**, 1629-1631, doi:10.1126/science.1158078 (2008).
- 7 Tajima, F. Statistical-Method for Testing the Neutral Mutation Hypothesis by DNA Polymorphism. *Genetics* **123**, 585-595 (1989).
- 8 Akashi, H. & Schaeffer, S. W. Natural selection and the frequency distributions of "silent" DNA polymorphism in *Drosophila*. *Genetics* **146**, 295-307 (1997).
- 9 Andolfatto, P. Adaptive evolution of non-coding DNA in *Drosophila*. *Nature* **437**, 1149-1152, doi:10.1038/nature04107 (2005).
- 10 Hahn, M. W. *Molecular population genetics*.
- 11 Williamson, S. H. *et al.* Simultaneous inference of selection and population growth from patterns of variation in the human genome. *Proceedings of the National Academy of Sciences of the United States of America* **102**, 7882-7887, doi:10.1073/pnas.0502300102 (2005).
- 12 Mackay, T. F. C. *et al.* The *Drosophila melanogaster* Genetic Reference Panel. *Nature* **482**, 173-178, doi:10.1038/nature10811 (2012).
- 13 Mackay, T. F. C. & Langley, C. H. Molecular and Phenotypic Variation in the Achaete-Scute Region of *Drosophila-Melanogaster*. *Nature* **348**, 64-66, doi:Doi 10.1038/348064a0 (1990).
- 14 Long, A. D., Lyman, R. F., Morgan, A. H., Langley, C. H. & Mackay, T. F. C. Both naturally occurring insertions of transposable elements and intermediate frequency polymorphisms at the achaete-scute complex are associated with variation in bristle number in *Drosophila melanogaster*. *Genetics* **154**, 1255-1269 (2000).
- 15 Schmidt, J. M. *et al.* Copy number variation and transposable elements feature in recent, ongoing adaptation at the *Cyp6g1* locus. *PLoS Genet* **6**, e1000998, doi:10.1371/journal.pgen.1000998 (2010).

REVIEWERS' COMMENTS:

Reviewer #1 (Remarks to the Author):

The authors have done a thorough job responding to the reviewer's comments, and I find the current version of the MS fully ready to publish. Not every suggestion made by reviewers was accommodated, but I find the justification reasonable. Most compelling is the last paragraph of the authors' rebuttal letter, which articulates clearly the value of the resource of these sequences and SV inferences, especially in reference to our understanding of the importance of SVs in complex trait variation.

Reviewer #2 (Remarks to the Author):

The authors have successfully addressed my concerns.